# Looming Angry Faces: Preliminary Evidence of Differential Electrophysiological Dynamics for Filtered Stimuli via Low and High Spatial Frequencies

**DOI:** 10.3390/brainsci14010098

**Published:** 2024-01-19

**Authors:** Zhou Yu, Eleanor Moses, Ada Kritikos, Alan J. Pegna

**Affiliations:** School of Psychology, The University of Queensland, Saint Lucia, Brisbane, QLD 4072, Australia; zhou.yu@uq.net.au (Z.Y.); eleanor.moses@uq.net.au (E.M.); a.kritikos@psy.uq.edu.au (A.K.)

**Keywords:** EEG/ERP, angry faces, looming, spatial frequencies, endogenous attention, P1, N170, P2

## Abstract

Looming motion interacts with threatening emotional cues in the initial stages of visual processing. However, the underlying neural networks are unclear. The current study investigated if the interactive effect of threat elicited by angry and looming faces is favoured by rapid, magnocellular neural pathways and if exogenous or endogenous attention influences such processing. Here, EEG/ERP techniques were used to explore the early ERP responses to moving emotional faces filtered for high spatial frequencies (HSF) and low spatial frequencies (LSF). Experiment 1 applied a passive-viewing paradigm, presenting filtered angry and neutral faces in static, approaching, or receding motions on a depth-cued background. In the second experiment, broadband faces (BSF) were included, and endogenous attention was directed to the expression of faces. Our main results showed that regardless of attentional control, P1 was enhanced by BSF angry faces, but neither HSF nor LSF faces drove the effect of facial expressions. Such findings indicate that looming motion and threatening expressions are integrated rapidly at the P1 level but that this processing relies neither on LSF nor on HSF information in isolation. The N170 was enhanced for BSF angry faces regardless of attention but was enhanced for LSF angry faces during passive viewing. These results suggest the involvement of a neural pathway reliant on LSF information at the N170 level. Taken together with previous reports from the literature, this may indicate the involvement of multiple parallel neural pathways during early visual processing of approaching emotional faces.

## 1. Introduction

A critical factor in survival is the efficiency of threat detection. Threats activate survival circuits and influence our behaviour, for example, by prompting approach or avoidance. As social creatures, emotional facial expressions convey significant threat cues that can prompt and inform these approach and avoidance responses [1,2,3,4]. Negative expressions like anger can motivate an avoidance response to escape threat and minimise conflict or harm [5,6,7,8]. Factors aside from expression, such as looming motion, also convey threats. We live in a dynamic world, and rapidly looming motion represents both a potential invasion of personal space and a potential collision [9,10,11,12,13,14]. As such, looming motion can elicit stereotypical reactions of fear [15,16,17], a response that appears to be apparent from birth [18,19,20,21] and monkeys [22].

Perhaps because of their evolutionary relevance, threat-relevant information from facial expressions can be differentiated at the early stages of visual processing. ERP studies reveal that emotional expressions evoke modulations that occur as early as 100 ms at the P1 over occipito-parietal sites [23,24,25]. A more robust finding is that the face-specific N170 is also emotion-sensitive [26,27,28], is typically enhanced by negative emotions like angry or fearful expressions for recent reviews [29,30], and is sometimes more pronounced in the right hemisphere [29,30]. Those early emotional effects suggest increased neural activation of facial expressions indicative of a threat.

Motion is also processed rapidly and has been found to modulate ERP markers such as the N1, P1, and P2. Motion onset or offset can evoke a posterior P1 component [31,32,33,34]. Looming motion is differentiated rapidly, with ERP effects reported at the P1 [35] and N1 [36]. Additionally, the P2 can reflect motion effects in response to stimuli saliency [37]. Therefore, motion and emotion are both processed rapidly and can signal threat-related information.

Interestingly, recent research has indicated that facial expressions and motion interact to modulate behavioural and neural responses, as shown by faster and more accurate responses for “looming” angry faces [1]. ERP studies have shed light on the neural mechanism underlying such occurrences. Yu et al. [37] found that angry expressions enhanced the P1 and N170 but that approaching angry expressions specifically prompted a further enhancement of the P1. This P1 enhancement could reflect the enhanced processing of threat-relevant faces [25,26,38,39]. Other schools of thought have suggested that such effects are the product of the processing of low-level visual information [40,41]. However, in a follow-up study, Yu et al. [42] replicated their looming emotional face study, including inverted faces. Face inversion is widely thought to destroy holistic processing and face/emotion recognition while maintaining low-level features [43,44,45,46,47,48,49,50,51]. The authors found that the enhanced P1 to looming angry expressions did not appear when faces were inverted, indicating that the early neural response required the identification of the emotional expression, thus corroborating the idea of rapid threat-related processing [42].

The early sensitivity to the interaction between motion and emotion suggests that the information converges rapidly through the different pathways. Regarding emotions, some research has proposed the existence of a rapid subcortical pathway for early threat detection. This subcortical path is purported to rely on projections from the retina to the superior colliculus and pulvinar and subsequently to the amygdala, allowing for the encoding of threats bypassing the typical thalamocortical streams [2,52,53,54,55,56]. This can be supported by case studies of individuals with blindness due to damage to the primary visual cortex, such as TN [57]. Although claiming nothing was seen, TN differentiated fearful facial expressions at a rate higher than chance; fMRI scanning also revealed that the right amygdala was activated during his performance [57,58]. With respect to motion processing, EEG and TMS studies in humans, as well as intracranial recordings in monkeys, have provided evidence that information regarding movement likely reaches cortical regions very rapidly (i.e., within ~50 ms) and is likely conveyed partly via magnocellular thalamo-extrastriate projections to V5 [34,59,60,61].

Interestingly, other studies have shown that when contrasted with receding stimuli, looming stimuli activate the superior colliculus and the pulvinar nucleus of the thalamus [62], as in the case of emotion and threat. For example, in the study by Cléry et al. [63] in marmosets, looming but not receding stimuli triggered strong and widespread activation in the superior colliculus and pulvinar areas and the putamen. Consequently, this suggests an overlap of the subcortical structures involved in the processing of looming motion and those involved in emotion. It is therefore reasonable to assume that these two types of information are integrated relatively early in time and that the initial interactive processing of looming motion and emotional expression may likely be conveyed through a rapid subcortical pathway directly to the amygdala, which relies on the superior colliculus and pulvinar.

One way to investigate the neural pathways underlying the processing of emotion and motion is to manipulate the spatial frequency content of the stimuli. Motion is processed via functionally specific aspects of the visual system. Magnocellular layers of the lateral geniculate nucleus encode large spatial regions at high temporal rates, processing low spatial frequency (LSF) at rapid speeds [64,65]. Indeed, behavioural and electrophysiological studies show faster responses to LSF information [66,67,68]. Meanwhile, parvocellular layers encode high spatial frequency (HSF) information, such as fine details of the stimuli over small spatial expanses and at a slower temporal rate [64,65]. Neural paths that receive input from these different layers are, therefore, sensitive to visual information at specific temporal resolutions and ranges of spatial frequencies [65,69,70].

While general face processing relies on input from a range of spatial frequencies, holistic and global processes like emotion perception have been suggested to rely on LSF information [71,72,73]. Structures involved in threat appraisal, like the amygdala, have been observed to activate LSF fearful faces [58,74], even when they are task-irrelevant [75]. Case studies where cortical lesions leave only the subcortical system intact show amygdala activation to emotional faces filtered to LSF but not HSF information [58]. Thus, the subcortical pathway is suggested to be reliant on LSF input. Furthermore, ERP studies support the idea that expression encoding is preferentially tuned to coarse visual input processed by magnocellular streams, with early modulations occurring for LSF-filtered faces. Indeed, larger posterior P1 amplitudes are evoked in LSF faces compared to unfiltered [76] and HSF faces [77,78].

This paper attempted to investigate the interaction of emotion and looming motion and the neural networks underlying these two types of threats. Given the posited sensitivity of magnocellular pathways to LSF information, manipulating spatial frequency was expected to provide insight into the involvement of the different pathways when conveying threat-related information. Using EEG/ERP techniques, the time course of neural responses to approaching and receding angry and neutral faces was assessed at varying levels of spatial frequencies (LSF, HSF, or unfiltered). In Experiment 1, dynamic LSF/HSF-filtered angry and neutral faces were compared with static ones in a passive viewing paradigm to rule out the initial influence of motion on effects relevant to spatial frequencies. Considering that endogenous attention might affect emotion modulations at the early stages [30], Experiment 2 required direct attention to the facial expressions and compared dynamic LSF-/HSF-filtered angry and neutral faces to unfiltered ones.

The study particularly explored the modulation pattern of emotion and motion on early ERP components of P1, N170, and P2 within each spatial frequency condition. We hypothesised that if looming motion and emotion rely on the magnocellular routes, approaching faces would enhance early responses, especially for LSF angry faces and that the effects of motion and emotion would interact. Due to the possible interference of low-level effects on the ERPs, different spatial frequency bands were not included in a single analysis, and effects were examined within each condition of spatial frequency separately. Crucially, we expected enhanced P1 and N170 responses for approaching angry faces in the LSF but not the HSF domain. In line with our previous observations, we predicted the P2 would show sensitivity to motion across spatial frequencies. Lastly, we expected endogenous attention to modulate emotion and motion effects on the early ERPs, especially filtered faces.

## 2. Experiment 1

### 2.1. Method

#### 2.1.1. Participants

Twenty-nine students from the University of Queensland participated in this experiment. After the exclusion of three participants due to noisy recordings, 26 participants (13 females) were retained (age: 17–33 years; M = 22.50; SD = 4.19). All participants had normal or corrected-to-normal vision and no self-reported neurological conditions. Participants were recruited via advertisement on campus; they received AUD 40, or two course credits, for participating in the study. Participation was voluntary, and the experiment only proceeded once participants signed the Consent Form. This study was approved by the University of Queensland Ethics Committee.

The sample size was sufficient according to our initial power analysis using MorePower 6.0.4 [79]. This study used a 2 × 2 × 3 within-subjects design (described below). With an alpha (α) of 0.05 and an optimal power (1 − β) of 0.8, the results indicated that a sample of 26 was sufficient to reflect significant two-level main effects and 2 × 2 interaction with a large effect size (ηp^2^ = 0.25); and was sufficient for other possible effects with an effect size of ηp^2^ = 0.17.

#### 2.1.2. Stimuli

The emotional faces of eight females and eight males were selected from the Radboud Faces Database [80], and the same identities were selected for their angry and neutral expressions. Original face pictures were first cropped to be squares of 512 × 512 pixels presenting the hair and face areas. Using a MATLAB 2022b [81] script developed by Perfetto et al. [82], each face was processed using the Butterworth2 filter in a low spatial frequency/LSF of <8 cpi (approx. 0.85 cpd) and a high spatial frequency/HSF of >32 cpi (approx. 3.4 cpd), respectively. Provided by the script, all the filtered pictures were also processed to have normalised contrast, thus maintaining the overall low-level visual features consistent across all of them. Each filtered face picture was imported to Gimp 2.0 (https://www.gimp.org, accessed on 4 July 2022) for elliptical cropping. The face area within a vertical ellipse of 285 × 375 pixels centred over the image middle point was visible, while the other pictorial area remained transparent (see Figure 1 for the example stimuli).

#### 2.1.3. Design and Procedure

The study used a 2 (Emotion: anger and neutral) × 2 (Filter: HSF and LSF) × 3 (Motion: approaching, receding, and static) within-participants design. All facial stimuli were presented on a full-screen depth-cued background throughout the experiment. The background consisted of black lines (RGB: 0 × 0 × 0) arranged as a polar projection on a grey screen (RGB: 50 × 50 × 50; see Figure 1). Participants were presented with faces that matched their sex to avoid potential gender biases.

Each trial started with a fixation cross in the screen centre for 1000 ms, followed by the appearance of a single upright face. In the static conditions, the face was presented at the screen centre for 500 ms with a constant size of 8.3° × 6.4° (H × W). In the approaching conditions, faces were presented initially at a size of 8.3° × 6.4° (H × W) and immediately expanded to 12.4° × 9.5° over 500 ms with a constant speed. In the receding conditions, the exact opposite motion was used, with faces appearing first at 8.3° × 6.4° and contracting to 4.2° × 3.2° over 500 ms. Following a random period between 600 and 1000 ms after the face offset, a random number between 1 and 9 appeared for 100 ms (see Figure 1 for a typical trial), followed by a blank screen that added up to 4 s of a complete trial. Participants were required to respond regarding whether the number was even (press the “E” key) or odd (press the “O” key) as fast as possible. This number categorisation task aimed to reinforce participants’ fixation on the screen centre. The experiment consisted of 10 blocks of 96 trials, with a break for up to 5 min between each block. All conditions were randomised within each block with an equal number of repetitions. The total participation time was, on average, 1.5 h.

#### 2.1.4. Apparatus

PsychoPy3 [83] was used to code and deliver the experiment. A screen with a resolution of 1080 × 1920 pixels and a refresh rate of 60 Hz (24-inch ASUS LCD monitor, model VG248QE, ASUSTeK Computer Inc. Taipei, Taiwan) placed 60 cm from the participants was used to present the stimuli. EEG data were recorded using the 64-channel BioSemi Active Two system (BioSemi Inc., Amsterdam, The Netherlands) at a sampling rate of 1024 Hz and a bandwidth (3 dB) of 208 Hz. Electrodes were positioned based on the extended international 10–20 system. Additionally, a Common Mode Sense (CMS) active electrode, coupled with a Driven Right Leg (DRL) passive electrode, functioned as the active reference and ground. These created a feedback loop designed to maintain the average potential similar to the reference voltage within the AD-box, essentially serving as the amplifier “zero” (https://www.biosemi.com/faq/cms&drl.htm, accessed on 16 January 2024).

#### 2.1.5. EEG Data Processing

EEG data were processed offline using BrainVision Analyzer (Version 2.2.0, Brain Products GmbH, Gilching, Germany). Data were downsampled to 512 Hz, recalculated against the average reference of all 64 electrodes, and filtered offline from 0.1 Hz to 30 Hz. For each participant, epochs of 100 ms pre-stimulus onset to 500 ms post-stimulus onset were used to compute the ERPs of interest, which were baseline-corrected using the 100 ms pre-stimulus period. Artefact screening was conducted on each epoch before averaging ERPs. We manually rejected epochs containing eyeblinks and bad traces that exceeded the ±60 µv threshold. Participants who had less than 40% of trials remaining in any condition (<32 trials per condition) were considered to have insufficient data and were excluded from further analyses. The participants included in the analyses had approximately 63 trials per condition on average (SD = 9.82).

Responses from all participants were averaged at each electrode to obtain the grand mean ERP traces. These traces were then used to generate the EEG topographic maps, thus, the visual identification of electrodes displaying the highest activity during the time window associated with each interested ERP component. We identified the Region of Interest (ROI) for each ERP component by selecting groups of electrodes that exhibited the highest activity during the specific time windows. These choices align with the common preferences found in the literature as well [84,85,86]. Then, for each participant, the average activity of electrodes within the ROI was calculated to generate a single ERP trace for each condition.

In this manner, one ROI was determined for the P1, which included nine electrodes at the posterior region (P7, P9, PO7, O1, Oz, O2, P8, P10, and PO8 electrodes). The grand mean ERP traces of the P1 showed their peak within 84–108 ms for each condition, which was consistent with the literature. Thus, a time window of 80–115 ms locked to the face onset was selected to compute the mean P1 values. Using a similar approach, two ROIs were selected for the N170, which included one on the left (TP7, P7, and P9) and one on the right (TP8, P8, and P10). The peak N170 was found between 146 and 166 ms for each condition. Consequently, the mean amplitude for the N170 was computed on a 30 ms time window between 140 and 170 ms and centred over the maximum. For the P2 component, one central ROI at the occipital site was created, including electrodes of O1, Oz, and O2. Mean amplitudes of the P2 were computed on a 40 ms window located between 200 and 240 ms and centred over the peaks.

### 2.2. Results

Mean amplitudes of each ERP were obtained as described in the Method section and were exported to JASP (Version 0.14.1) for statistical analysis. The descriptive statistics (mean and standard deviation) of the ERP amplitudes of each condition can be seen in the Appendix A. Topographic maps and grand traces of ERP of interest can be seen in Figure 2, Figure 3, Figure 4 and Figure 5. For ease of readability, the ERP traces of the N170 and P2 presented collapsed conditions reflecting significant main or interactive effects. Comprehensive ERP traces can be found in the Appendix A. A series of repeated measures ANOVAs on the amplitudes were performed at each level of spatial frequency condition for each ERP component, respectively.

#### 2.2.1. P1 Component

At both LSF and HSF levels, a 2-way repeated measures ANOVA was performed for Emotion (angry and neutral) and Motion Direction (approaching, receding, and static), respectively.

For HSF faces, there was no main effect of facial expression: F (1, 25) = 0.48, *p* = 0.497, η_p_^2^ = 0.019; or motion direction, F (2, 50) = 0.16, *p* = 0.851, η_p_^2^ = 0.006. No interaction was found either: F (2, 50) = 0.25, *p* = 0.779, η_p_^2^ = 0.010.

Similarly, for LSF faces, there were no main effects of facial expression: F (1, 25) = 0.36, *p* = 0.555, η_p_^2^ = 0.014; motion, F (2, 50) = 0.07, *p* = 0.928, η_p_^2^ = 0.003; or interaction found significant either, F (2, 50) = 1.33, *p* = 0.275, η_p_^2^ = 0.050.

**Figure 2 brainsci-14-00098-f002:**
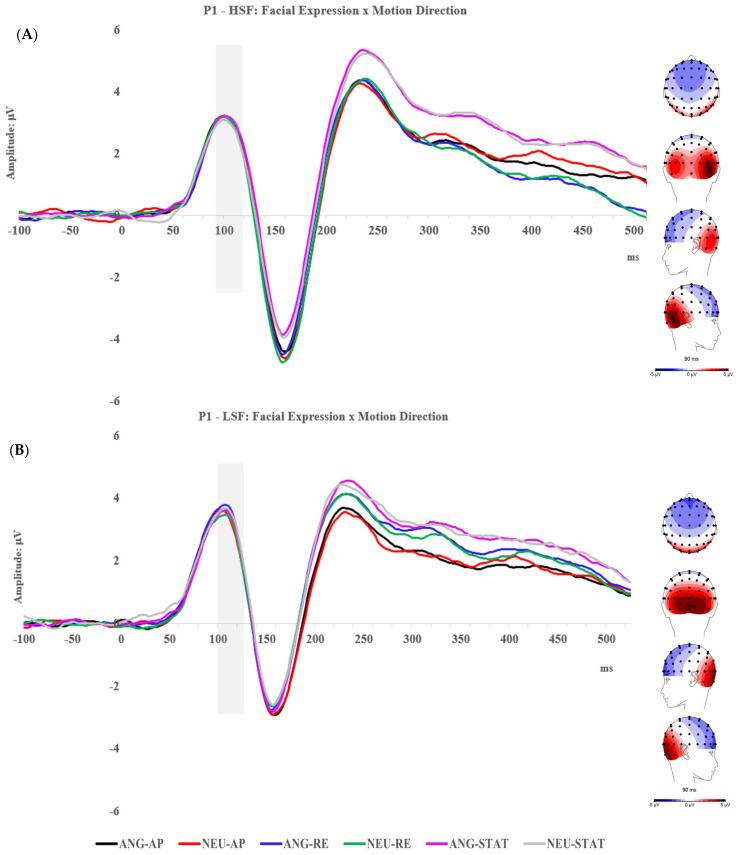
Topographic maps and grand ERP traces of each condition for the P1, separately presented by HSF (**A**) and LSF (**B**) conditions. AP: approaching; RE: receding; ST: static; ANG: angry; NEU: neutral.

#### 2.2.2. N170 Component

At each level of spatial frequency condition (LSF vs. HSF), a 3-way repeated measures ANOVA was performed for ROI (L and R), Emotion (angry and neutral), and Motion Direction (approaching, receding, and static). Only a Motion main effect was found for HSF faces, F (2, 50) = 14.13, *p* < 0.001, η_p_^2^ = 0.361. Using Bonferroni correction, the post hoc comparisons showed that the N170 was significantly enhanced by approaching and receding motions when each was compared with the static condition; t_approach_ (25) = 3.93, *p_bonf_* = 0.002, t_reced_ (25) = 5.43, *p_bonf_* < 0.001. However, no difference between the two moving conditions was found; t (25) = 1.08, p_bonf_ = 0.867. In sum, stronger responses for moving than static stimuli are found at the N170 for HSF information: receding (−4.15 μV) = approaching (−4.02 μV) > static (−3.59 μV).

**Figure 3 brainsci-14-00098-f003:**
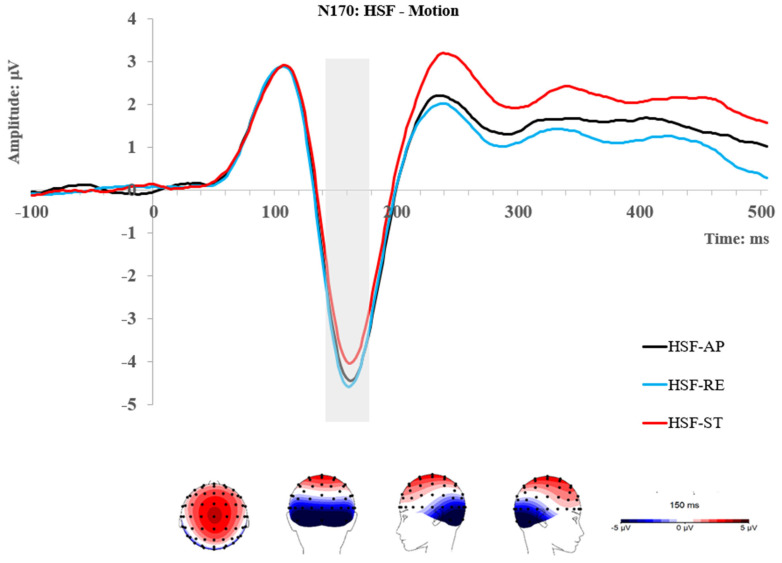
Topographic maps and grand ERP traces for the HSF conditions; the traces present the significant main effect of motion at the N170. L: left ROI; R: right ROI; AP: approaching; RE: receding; ST: static.

For the LSF face, a significant main effect of motion was also found; F (2, 50) = 4.31, *p* = 0.019, η_p_^2^ = 0.147. Post hoc comparisons showed that the N170 was significantly enhanced by approaching when compared with receding motion; t (25) = 2.65, *p_bonf_* = 0.042. However, no other comparisons reached significance. Showing an overall pattern of approaching~static (−3.42 μV) = receding (−3.38 μV), approaching (−3.72 μV) > receding. Furthermore, an interaction between ROI and emotion was found; F (1, 25) = 4.74, *p* = 0.039, η_p_^2^ = 0.159. A follow-up Simple Main Effect analysis showed that the N170 was enhanced by anger compared with neutral faces at the right ROI; F (1, 25) = 4.58, *p* = 0.042, η_p_^2^ = 0.155; however, no effect of emotion was found at the left ROI; F (1, 25) = 0.18, *p* = 0.736, η_p_^2^ = 0.007. No other main or interactive effect was found on the N170 either.

**Figure 4 brainsci-14-00098-f004:**
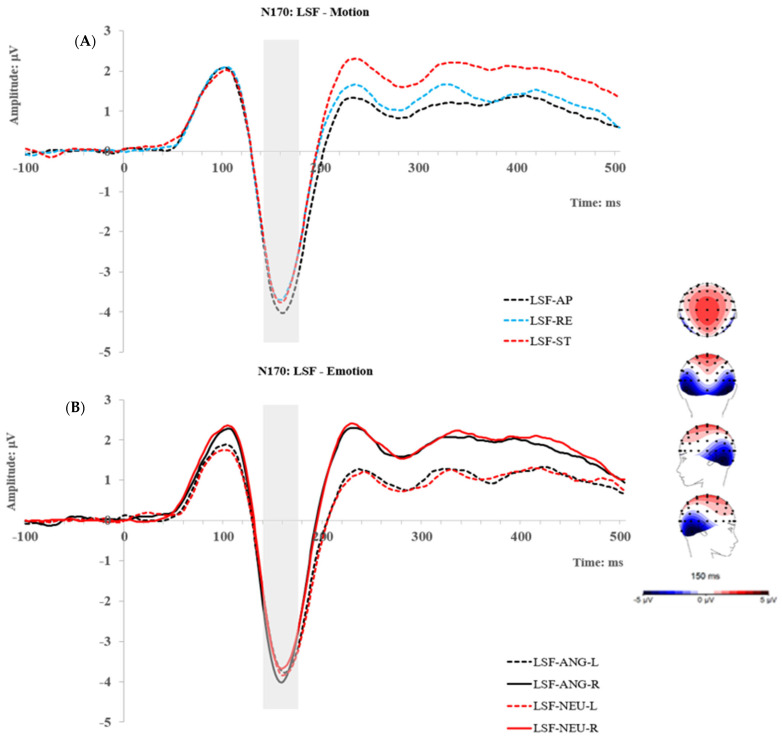
Topographic maps and grand ERP traces for the LSF conditions, the traces present the significant main effect of motion (**A**) and the interactive effect of emotion and ROI (**B**) at the N170. ANG: angry; NEU: neutral; L: left ROI; R: right ROI; AP: approaching; RE: receding; ST: static.

#### 2.2.3. P2 Component

At the HSF level, a 2 (angry and neutral) × 3 (approaching, receding, and static) repeated measures ANOVA was performed. The main effect of Motion reached significance; F (2, 50) = 5.35, *p* = 0.008, *p*_Huynh-Feldt corrected_ = 0.017, η_p_^2^ = 0.176. Post hoc comparisons using Bonferroni correction revealed significantly smaller activity in approaching than receding and static conditions; t (25) = −3.27, *p_bonf_* = 0.009, t (25) = −3.15, *p_bonf_* = 0.013, respectively. No difference between receding and static conditions was shown. Overall, neural activities at the P2 show a pattern of approaching (4.04 μV) < receding (4.6 μV)~static (4.95 μV).

At the LSF level, the same analysis of ANOVA as HSF was used. It also showed a main effect of motion; F (2, 50) = 10.76, *p* < 0.001, η_p_^2^ = 0.301. Interestingly, post hoc comparisons also revealed the same pattern of results as the HSF condition, such that the P2 was significantly lower for approaching faces (4.69 μV) than receding (5.41 μV); t (25) = −3.81, *p_bonf_* = 0.002, and static faces (5.33 μV), t (25) = −4.22, *p_bonf_* < 0.001, with no difference between receding and static faces. No other main or interactive effect was found on the P2.

**Figure 5 brainsci-14-00098-f005:**
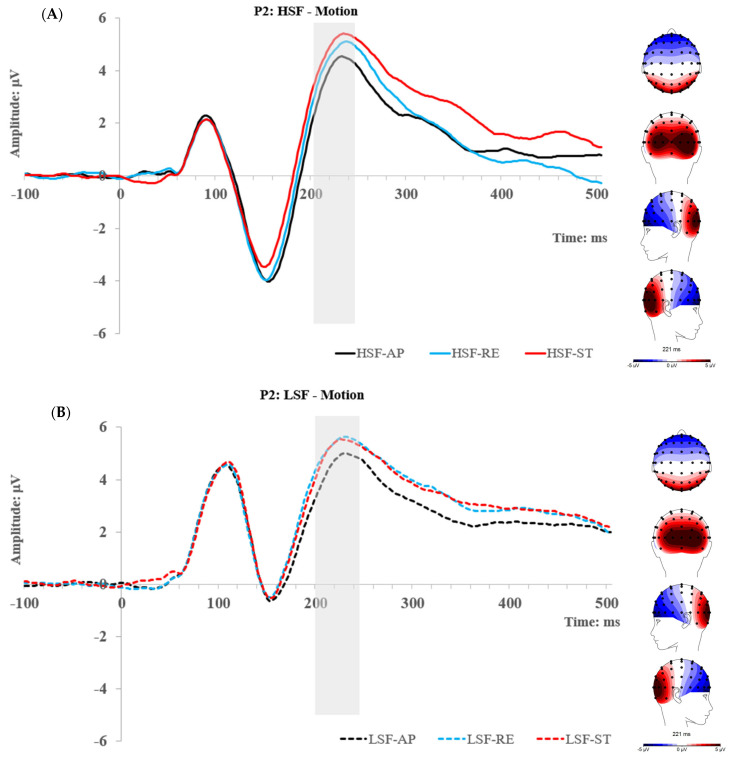
Topographic maps and grand ERP traces for the HSF (**A**) and LSF conditions (**B**), respectively, and the traces present the significant main effect of motion at the P2. AP: approaching; RE: receding; ST: static.

### 2.3. Experiment 1 Summary

Using a passive-viewing paradigm, Experiment 1 aimed to investigate the P1, N170, and P2 modulation of facial expressions and motion via HSF- and LSF-filtered faces. At the P1 stage, neither HSF nor LSF faces evoked effects associated with facial expression or motion. The N170 was enhanced by LSF angry faces, although the effect was only significant in the right hemisphere. Interestingly, the N170 also reflected differential motion effects evoked by HSF and LSF groups. The HSF conditions reflected a differentiation between moving (i.e., approaching and receding) versus static status, while among the LSF conditions, approaching motion was differentiated from the others. The P2 demonstrated overall sensitivity to motion. Across the HSF and LSF conditions, the approaching motion was differentiated from the others (i.e., receding and static) by showing smaller amplitudes.

## 3. Experiment 2

Consistent with our initial expectation, the N170 in Experiment 1 showed enhancement to angry faces filtered by LSF. This result supports our hypothesis that coarse processing via magnocellular routes is involved in the early processing of threatening faces. The P1 was not modulated by moving compared to static faces, suggesting a minimum influence of motion on initial effects relevant to spatial frequencies. Surprisingly, the emotional modulations for filtered faces were not found at the P1. It suggests that the high or low spatial frequency bands alone are insufficient for early facial expression processing during passive viewing. Indeed, some studies showed that the neural processing of emotional faces requires attentional resources [87,88]. Attentional focus and task demand tend to modulate the effects of emotional expressions on early ERPs [30]. Thus, an alternative explanation to our P1 results is that faces filtered by HSF or LSF are not sufficiently attended during passive viewing, thereby not meeting the sensory processing threshold. To investigate this interpretation, Experiment 2 followed up with a task of facial expression discrimination on LSF, HSF, and unfiltered faces, thus requiring endogenous attention directed to the emotional expressions. Since processes associated with voluntary attentional control recruit cortical pathways [89,90,91], Experiment 2 can further help in interpreting the processing as being subcortical or cortical. If endogenous attention could engage the processing of emotional expressions, we expected that the P1 and N170 would be enhanced for approaching angry faces in LSF and unfiltered conditions, and the P2 would only reflect sensitivity to motion direction. If cortically based processing were involved in the early stages, we would expect the ERPs to show different modulations of facial expressions and motion between the passive and active viewing tasks. However, we are open to observing the potential modulation by endogenous attention.

### 3.1. Method

#### 3.1.1. Participants

Following the same recruitment and reimbursement procedure described in Experiment 1, 33 participants volunteered in this experiment. After removing three participants who had insufficient data or noisy signals, 30 participants (20 females and 10 males) aged 19–34 (M = 24.23; SD = 3.46) were included for subsequent analysis. All participants had normal or corrected-to-normal vision and no self-reported neurological condition. The sample size is also considered sufficient, given it is larger than what was in the previous experiment.

#### 3.1.2. Design and Procedure

The same HSF and LSF facial stimuli as in Experiment 1 and their unfiltered version (Broadband Spatial Frequencies/BSF) were used for this experiment (Figure 6). The contrast was normalised to maintain the same low-level visual features across all faces. The static condition was excluded based on the findings of the previous experiment, which demonstrated no influence of motion on effects relevant to spatial frequency initially (i.e., on the P1). Moreover, it could make the current experiment more manageable. Thus, this experiment used a 2 (Emotion: angry and neutral) × 2 (Motion Direction: approaching and receding) × 3 (Filter: LSF, HSF, and unfiltered) within-participants design. The depth-cued background was also displayed along with every face presentation.

Instead of number categorisation as an irrelevant task, this experiment employed an emotion identification task and asked participants to pay explicit attention to the expression of each face. Each trial started with a fixation cross in the centre of the screen for 1000 ms, followed by the appearance of a single face, which started approaching or receding immediately upon onset. The approaching or receding rates are the same as in Experiment 1. Thus, in the approaching conditions, faces were presented initially at a size of 8.3° × 6.4° (H × W) and immediately expanding to 10.4° × 7.9° over 250 ms with a constant speed. In the receding conditions, the exact opposite motion was used, with faces appearing first at 8.3° × 6.4° and contracting to 6.3° × 4.8° over 250 ms. After 750 ms of a blank screen following the face offset, a question of either “Was the face angry?” or “Was the face neutral?” appeared on the screen. Participants were instructed to respond “yes” (press the “Y” key) or “no” (press the “N” key). They were also informed that the response was not timed and that they only needed to be as accurate as possible. The question remained the same for each block (e.g., only ask about being angry), but the order of questions was randomised across blocks with equal repetition. This was meant to maintain the participants’ level of engagement constant throughout. The experiment consisted of 10 blocks of 96 trials with a break for up to 5 min between each block. All conditions of the facial stimuli were randomised within each block with an equal number of repetitions. The total participation time was, on average, 1.5 h.

#### 3.1.3. Apparatus and EEG Data Processing

EEG data were acquired using the same set up described in Experiment 1. Filtering and artefact screening/rejection criteria identical to Experiment 1 were also applied. Participants included in the analyses had approximately 65 trials per condition on average (SD = 8.38).

Following the same approach of identifying ROIs as in Experiment 1, for each ERP component of interest, the electrodes included for each ROI were consistent with the previous experiment. Thus, one ROI was determined for the P1 at the posterior site (O1, Oz, O2, P7, P9, PO7, P8, P10, and PO8 electrodes). A time window of 85–115 ms locked to the face onset was selected to compute the mean P1 values. For the N170, one ROI on the left (TP7, P7, and P9) and one on the right (TP8, P8, and P10) were included. The mean amplitude for the N170 was computed on a 40 ms time window between 135 and 175 ms and centred over the maximum. For the P2, one central ROI was created at the occipital site, including electrodes of O1, Oz, and O2. Mean amplitudes of the P2 were computed on a 40 ms window located between 190 and 230 ms and centred over the peaks.

### 3.2. Results

As described above, the mean amplitudes of each ERP were obtained and exported to JASP (Version 0.14.1) for statistical analysis. The descriptive statistics (mean and standard deviation) of the ERP amplitudes can be seen in the Appendix A. Topographic maps and grand traces of each ERP of interest are visually alike to those in Experiment 1; thus, they are not presented again. However, comprehensive ERP traces can be found in the Appendix A. Using a similar analysis approach as in Experiment 1, repeated measures ANOVAs were performed.

#### 3.2.1. P1 Component

Three separate 2-way repeated measures ANOVA were performed for Emotion (angry and neutral) and Motion Direction (approaching and receding) at SF, HSF, and LSF conditions, respectively.

For unfiltered/BSF faces, a significant main effect of motion was found; F (1, 29) = 9.84, *p* = 0.004, η_p_^2^ = 0.253, such that an overall enhanced P1 was found for receding (3.94 μV) when compared with approaching (3.61 μV) motion. A significant main effect of emotion was also found, F (1, 29) = 14.15, *p* < 0.001, η_p_^2^ = 0.328, showing that the P1 was overall stronger for angry (3.95 μV) than neutral (3.60 μV) faces.

For HSF faces, there were no significant main effects of facial expression, motion, or interaction at all.

For LSF faces, it also revealed a main effect of motion; F (1, 29) = 5.39, *p* = 0.027, η_p_^2^ = 0.157. Like the SF condition, it also showed an enhanced P1 for receding (4.01 μV) compared to approaching motion (3.76 μV).

#### 3.2.2. N170 Component

At BSF, LSF, and HSF, a 2 ROI (L and R) × 2 Emotion (angry and neutral) × 2 Motion (approaching and receding) repeated measures ANOVA was performed, respectively. For BSF faces, a main effect of emotion was found, F (1, 29) = 4.40, *p* = 0.045, η_p_^2^ = 0.132, such that the N170 was overall enhanced for angry (−4.33 μV) compared with neutral (−4.05 μV) faces. Furthermore, an interaction between ROI and emotion was found; F (1, 29) = 4.22, *p* = 0.049, η_p_^2^ = 0.127. A follow-up Simple Main Effect analysis revealed that the N170 was significantly enhanced by angry faces at the right ROI, F (1, 29) = 9.42, *p* = 0.005, η_p_^2^ = 0.245, but no difference at the left ROI, F (1, 29) = 0.37, *p* = 0.547, η_p_^2^ = 0.013. This indicates that the N170 emotion effect for SF faces is mainly explained by activities in the right ROI.

For HSF faces, only a Motion main effect was found, F (1, 29) = 5.91, *p* = 0.021, η_p_^2^ = 0.169, where the N170 was overall stronger for receding (−4.19 μV) than approaching (−3.93 μV) faces. Interestingly, no main or interactive effect was found for LSF faces at the N170.

#### 3.2.3. P2 Component

At each level of spatial frequency, a 2 (angry and neutral) × 2 (approaching and receding) repeated measures ANOVA was performed, respectively. For all conditions of spatial frequency, main effects of Motion were found, although they were marginally significant for BSF condition; F_BSF_ (1, 29) = 3.96, *p* = 0.056, η_p_^2^ = 0.12, F_HSF_ (1, 29) = 8.53, *p* = 0.007, η_p_^2^ = 0.227, F_LSF_ (1, 29) = 11.88, *p* = 0.002, η_p_^2^ = 0.291. Consistently, the same pattern of motion effect was found across all the SF conditions, such as the P2 tended to be more enhanced by receding than approaching motions (Table 1). No other main or interactive effect was found on the P2.

### 3.3. Experiment 2 Summary

Experiment 2 aimed to investigate how endogenous attention would modulate early components of looming emotional faces within different spatial frequency bands. Results on the P1 and N170 revealed overall enhancement to unfiltered angry faces, consistent with our previous studies [37,42]. As in Experiment 1, neither HSF nor LSF angry faces modulated the P1. Surprisingly, the P1 was overall enhanced by receding motion for both unfiltered and LSF groups, and the N170 showed no enhancement to LSF angry faces with endogenous attention. The motion effect for LSF conditions was not found either, although a general enhancement to HSF receding faces was shown. Lastly, the P2 demonstrated consistent results by always showing smaller amplitudes for approaching compared to receding faces across all spatial frequency groups.

## 4. Discussion

The current study used EEG/ERP to investigate the time course of neural activation to apparently approaching and receding emotional faces filtered by low and high spatial frequency bands. It aimed to determine if and when looming motion and facial threat interacted and whether this processing involved the magnocellular neural pathways. This latter point was examined by presenting low vs. high spatial frequency components of the stimuli, under the expectation that this would preferentially activate magno- vs. parvocellular pathways, respectively. The effect of endogenous attention on these neural events was also investigated. Experiment 1 used a passive-viewing paradigm as in our previous research. It presented HSF- and LSF-filtered angry and neutral faces in static, approaching, or receding motions on a depth-cued background. In the second experiment, endogenous attention was engaged by directing attention to the facial expression. The same HSF and LSF faces and their unfiltered/broadband counterparts (BSF) were used, presented in either approaching or receding motion.

### 4.1. The P1 Component

In this study, the early modulations of threat-relevant stimuli were hypothesised to rely on rapid and coarse processing via magnocellular routes. Our previous passive-viewing studies with unfiltered looming angry faces showed a P1 enhancement to angry faces that were further boosted by looming motion. We expected to observe similar P1 effects for LSF-filtered looming angry faces as had been seen for the unfiltered stimuli in our previous studies. However, contrary to our expectations, across both experiments of this study, faces filtered for LSF or HSF were not found to modulate the P1 responses to facial expressions, nor did they elicit interaction with looming motion.

The absence of any threat-relevant effect on our P1 for filtered faces is inconsistent with studies showing a stronger neural sensitivity to LSF-threatening faces at this stage [92,93,94,95,96]. One possible interpretation for this result pointed to the difference in task relevance of the faces. The null results of spatial frequency in our study were observed when passively viewing the faces, and studies reporting LSF-threat sensitivity at the P1 stage frequently required directed attention to the faces [92,93,95,96]. In line with this interpretation, several studies have demonstrated that the neural processing of emotional faces requires attentional resources [87,88]. Therefore, we followed up with Experiment 2, in which endogenous attention was directed to the facial expressions. We further hypothesised that attentional focus might enhance the initial visual processing of facial expressions, and it could be reflected in the P1 modulation by LSF emotional faces. Nevertheless, the P1 was again not modulated by either LSF- or HSF-filtered faces. Together, our results suggest that the P1 differentiation of threatening faces is not driven by a neural pathway that relies on coarse/LSF information alone.

On the other hand, the overall P1 differentiation of facial expressions for unfiltered faces was observed regardless of endogenous attention. It seems therefore possible that by filtering a face, LSF or HSF might remove the visual information crucial for expression-related processing at this stage or could reduce the intensity of sensory input required to achieve the processing threshold. As a result, only the unfiltered faces in our studies presented sufficient sensory information for the differentiation to be reflected at the P1. More importantly, our P1 enhancement for unfiltered angry faces is consistent with studies reporting its sensitivity for threat-related faces [25,26,38,39]. The timing of this effect is in line with the involvement of a rapidly activating neural pathway, which has been hypothesised to occur via the amygdala in early threat detection [95,96,97]. A recent iEEG study has reported stronger amygdala activities for BSF and LSF fearful faces beginning 74 ms post-stimulus onset, earlier than the fear-related response measured at the visual cortex [96]. Wang et al. [95] also reported amygdala activation to subliminal fearful faces within a window of 45–118 ms post-onset using iEEG methods. Moreover, patient studies of amygdala damage have demonstrated a loss of P1 modulations for threatening expressions [97]. Our findings thus keep with the literature suggesting rapid emotional threat processing. However, since we observed no emotional modulations of LSF faces on the P1, we cannot conclude an involvement of a magnocellular pathway. Consequently, we cannot determine if looming motion and emotion interact at the level of subcortical structures such as the superior colliculus or the pulvinar.

The other important result was that endogenous attention impacted the motion effects of the P1. There seemed to be a dissociation between passive and active viewing for the P1 responses of motion direction. The P1 only showed looming interactions with angry expressions among unfiltered faces when viewed passively. When directing endogenous attention, instead of the interaction in the looming condition, an overall P1 enhancement by receding motion was observed for both unfiltered and LSF faces. This indicates that spatial frequency filtering and endogenous attention differentially influence the initial processing of facial expressions and motion.

It appeared that during passive viewing, modulation via looming motion during the P1 relied on the perception of facial expressions; filtering attenuates P1 modulation by emotional expressions, and looming motion does not further interact with angry expressions. This finding aligns with our study using inverted faces [42]—the P1 showed no differentiation of angry faces nor further interaction with the looming motion to inverted faces. Since it is known that inversion impairs early recognition of faces and facial expressions, we took the P1 results to indicate that further enhancement by looming motion is dependent on the initial perception of facial expressions [42].

Endogenous attention, on the other hand, might enhance motion processing. The P1 enhancement to receding motion was observed for LSF-filtered faces, suggesting an increased ability to differentiate motion direction via the magnocellular channels. Alternatively, endogenous attention might activate the processing of some facial aspects that happen to be associated with motion direction. For example, face recognition might be improved with receding motion. Receding corresponds to a decrease in the retinal size of the stimuli. Thus, receding faces should proportionately include more facial features inside the foveal visual field, resulting in better recognition [64,65,98]. Further to this, it was found that LSF face images were more recognisable when presented at the size of a 2° visual angle compared with a 10° of visual angle [99]. The discrimination task indeed implied a higher requirement for face recognition. The enhanced P1 to faces moving away may thus have derived from an increase in neural activity linked to improved face recognition during receding motion, which is also applicable to LSF faces in this context.

The above interpretation further suggests that the initial processing of facial features is highly integrated with their dynamic aspects, and attentional properties modulate this processing. One could hypothesise, therefore, that automatic threat detection may occur in the context of exogenous attention via recruitment of a rapid subcortical-based neural pathway, but when endogenous attention is explicitly required by the task, visual processing of the task-relevant aspects gain priority, leading to the recruitment of higher-level computation networks for facial recognition which are improved with receding motion. Nevertheless, the receding effect could also reflect a more general sensitivity present for all objects with focused attention. To clarify this, a direct comparison between faces and other objects across spatial frequency conditions and more specific neuroimaging and analysis methods are required.

### 4.2. The N170 Component

As consistently reported in our previous studies, unfiltered angry faces globally enhanced the N170. The processes underlying this effect were previously hypothesised to be linked to rapid and coarse processing via magnocellular routes [64,65]. We therefore expected to observe enhanced N170 to both BSF and LSF angry faces across attentional conditions. We found that the N170 enhancement of BSF angry faces occurred under endogenous attention conditions as well. However, LSF angry faces only enhanced the N170 during passive viewing.

Our results showed that the N170 sensitivity to threatening facial expressions can be conveyed by LSF information under exogenous attention. This supported our hypothesis regarding the involvement of a magnocellular-based neural pathway in the N170 effect. However, our results also showed endogenous attention to facial expressions disrupts this N170 sensitivity for LSF input. Thus, the processing of LSF faces appears vulnerable to mechanisms associated with endogenous attention. This appears to suggest that the pathway for LSF input underlying the N170 may be cortical. This would be supported by some fMRI and TMS data, which suggest a dissociable cortical pathway between dynamic and static faces, with a crucial role of the superior temporal sulcus (STS) in response to moving faces [100,101]. Integration of recent literature shows that this “dynamic-face” pathway projects from the early visual cortex via motion-selective areas (V5/MT) into the STS [102]. The direct functional connection for motion perception between V1/V2 and V5/MT has been largely described [65,103,104], and the earliest activation of these areas has been reported to be within 30–50 ms following motion onset [59]. Another line of evidence reported that cortical regions, including the ventral prefrontal cortex (vPFC) and the insula, are also sensitive to emotional stimuli [105,106], and early differentiation in these areas can be achieved within 120–140 [105,107,108].

According to the above, it is possible that the processing of motion and facial expressions are integrated along this cortical “dynamic-face” pathway and lead to the modulation observed in the N170 window. Given that the rapid projection of motion input to the cortex (i.e., V5/MT) occurs mainly via magnocellular channels [64,65], it is possible that the LSF information regarding moving angry faces may be conveyed via this pathway to V5/MT and STS, although this interpretation of our current findings remains purely speculative. Regardless, the neural pathways involved in the processing of dynamic faces are likely modulated by endogenous attention. For example, using fMRI and diffusion tractography techniques, a recent study has demonstrated significant attentional modulatory activity along the posterior inferotemporal cortex parietal and frontal attentional regions [109]. This provides structural evidence for our interpretation that voluntary attentional control is necessary for emotional differentiation to LSF faces at the N170.

Importantly, our results showed that directed endogenous attention did not attenuate N170 enhancement for BSF angry faces, suggesting that the broad range of visual information contained in the unfiltered faces may be less sensitive to decreased attentional control. Moreover, the involvement of multiple neural pathways in the early differentiation of emotional faces [108,110] may produce a greater robustness to variations in endogenous attention.

Interestingly, motion directions tended to modulate the N170 when faces were filtered in HSF or LSF bands. These modulations varied across filtering and attentional conditions. These results are unexpected and difficult to interpret. Arguably, they may be the result of multiple parallel pathways being activated for different characteristics of the stimuli. However, this question would require additional investigations to be clarified.

### 4.3. The P2 Component

According to our previous study [37], we expected to see only motion sensitivity at the P2. As a negative correlation between the P2 amplitude and stimulus saliency has been reported [111] and the saliency of a visual stimulus increases with its pictorial size [112], we expected to find a smaller P2 for the approaching condition. Our results were as predicted across spatial frequency and attention conditions. Those results align with the literature, supporting that approaching stimuli are more salient, potentially due to increased sizes. We showed that this saliency effect of motion is independent of spatial frequency, further confirming the sensitivity to motion irrespective of the stimulus content. Although the P2 amplitudes to HSF faces followed a pattern of approaching < receding < static conditions when viewed passively, only the difference between approaching and static faces was significant. Endogenous attention tended to enhance differentiation between approaching and receding motions in HSF. This may be because HSF conveys fine details of the stimuli, and size changes associated with those details would be better processed with engaged attentional focus. This would align with findings showing engaged attentional effects at the P2 stage [24,113,114,115].

## 5. Conclusions

This study aimed to investigate the potential involvement of magnocellular pathways, which are thought to involve preferentially LSF inputs, on looming angry faces and further explored the effect of endogenous attention on neural responses. In two experiments, we used EEG/ERP techniques to measure the P1, N170, and P2 components in responses to approaching or receding emotional faces filtered by low and high spatial frequency bands under passive viewing and directed attention tasks. Our findings indicated that P1 was enhanced by BSF angry faces regardless of attentional control, while HSF and LSF faces did not elicit this effect. This suggests that looming motion and threatening expressions interact at the level of the P1. However, this effect does not rely on LSF or HSF inputs in isolation. The N170 showed an enhanced response to BSF angry faces irrespective of endogenous attention, but this enhancement was only evident during passive viewing. This is tentatively interpreted as indicating the involvement of a cortical neural pathway underlying the N170, differentiating LSF facial expressions. Overall, the results provided preliminary support for the involvement of multiple parallel neural pathways in the processing of looming emotional faces, with spatial frequency filtering and attentional control producing differential effects on these components.

## Figures and Tables

**Figure 1 brainsci-14-00098-f001:**
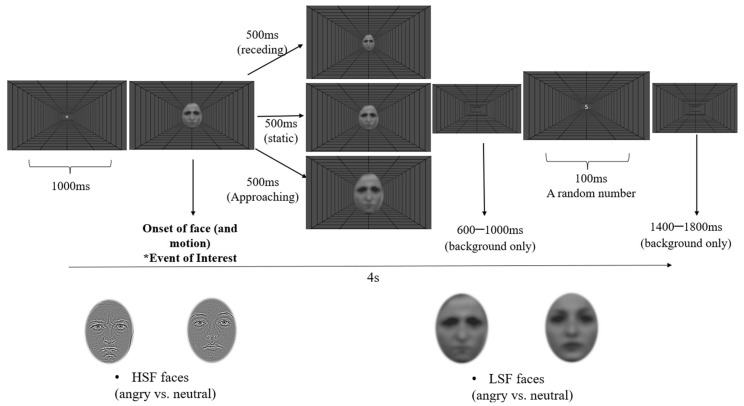
A typical trial procedure, with examples of facial stimuli in each condition. * Event of Interest marks the 0 ms of each epoch in the analysis.

**Figure 6 brainsci-14-00098-f006:**
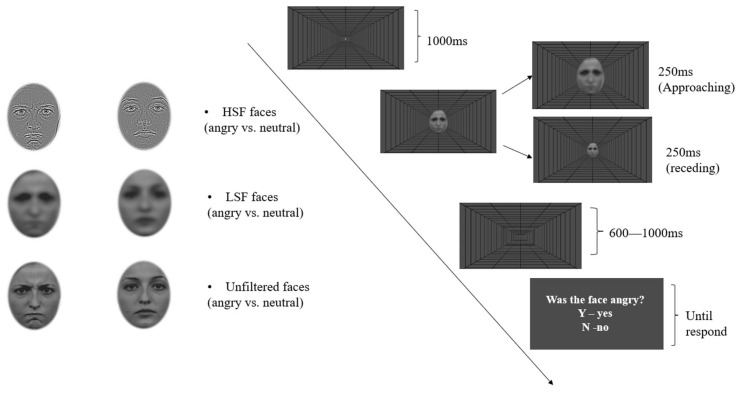
A typical trial procedure, with examples of facial stimuli in each condition.

**Table 1 brainsci-14-00098-t001:** P2 Amplitudes—Motion at each SF band (Marginal Means μV).

	BSF	LSF	HSF
Approaching	3.809	4.246	0.466
Receding	4.236	5.105	1.230

## Data Availability

Data collected for this research are stored confidentially on UQ Research Data Manager (https://research.uq.edu.au/rmbt/uqrdm, accessed on 23 October 2023); they are available upon request.

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
