# Peer review of "Looming Angry Faces: Preliminary Evidence of Differential Electrophysiological Dynamics for Filtered Stimuli via Low and High Spatial Frequencies"

_brainsci, 2024, doi:10.3390/brainsci14010098_

Round 1

Reviewer 1 Report

Comments and Suggestions for Authors

In the current study Yu and colleagues used EEG to determine “if the interactive threat effect produced by looming angry faces” reported in previous studies “is a product of a rapidly activating, coarse subcortical pathway”. To test this hypothesis, they decomposed face images in their LSF and HSF components. The idea is that, since signals from subcortical pathways rely on coarse information tuned to low spatial frequencies, the enhanced P1 response previously reported for angry looming faces should be evident over LSF stimuli and not on HSF ones. No evidence of such relationship was found as neither LSF nor HSF elicited an enhanced P1 response. Authors only confirmed (in Exp 2) previous results of larger responses for angry faces in BSF stimuli.

Although the topic of the study is interesting, I don’t consider it suitable for publication in Brain Sciences. Here are some of the motivations that led me to this choice alongside with some suggestions.

·           My main concern is about the design of the study which I consider not entirely suitable to answer the main question. I don’t see how authors could rule out the influence of the plethora of factors known to affect P1 amplitude while saying something very specific on the recruitment (or not) of visual subcortical routes. They conclude that their results suggest the “Existence of a rapid pathway for differentiating emotional expressions at the P1, but that this pathway is not reliant on LSF information”. First, the term “rapid” is ambiguous as seems to suggest the involvement of pathways other than the primary visual route. Second, this conclusion per se is also ambiguous as it should rather state “that this pathway is not reliant on LSF and HSF information in isolation” as authors found a null result even when using HSF stimuli.

·           The analysis of motion signals conveyed by subcortical routes are expected to reach the cortex at earlier latencies than those here explored (~50 ms). This has been extensively reported both on intracranial recording studies on monkeys (see for example Kawano et al., 1994), TMS studies (see for example Grasso et al. 2018; Laycock et al., 2007) and EEG studies on humans (see for example Ffytche et al., 1995). In the light of such literature, how would the authors claim that the enhanced P1 and N170 ERP responses reflect the recruitment of fast subcortical routes even in the case of a positive result on LSF stimuli?

Cited References

Kawano, K., Shidara, M., Watanabe, Y., & Yamane, S. (1994). Neural activity in cortical area MST of alert monkey during ocular following responses. Journal of neurophysiology71(6), 2305-2324.

Grasso, P. A., Làdavas, E., Bertini, C., Caltabiano, S., Thut, G., & Morand, S. (2018). Decoupling of early V5 motion processing from visual awareness: A matter of velocity as revealed by transcranial magnetic stimulation. Journal of cognitive neuroscience30(10), 1517-1531.

Laycock, R., Crewther, D. P., Fitzgerald, P. B., & Crewther, S. G. (2007). Evidence for fast signals and later processing in human V1/V2 and V5/MT+: a TMS study of motion perception. Journal of Neurophysiology98(3), 1253-1262.

Ffytche, D. H., Guy, C. N., & Zeki, S. (1995). The parallel visual motion inputs into areas V1 and V5 of human cerebral cortex. Brain118(6), 1375-1394.

·           I am not convinced about the statistical analyses used. In particular, different ERP components were analyzed using ANOVAs with a different number of categories within the level of ROI. In my view, it would be more straightforward to average across the n electrodes eliciting the largest response regardless of their scalp localization. This would have reduced the number of levels while also making ANOVAs more comparable to each other. In addition, it would have also made ERPs plots much more comprehensible. Now ERPs are, sometimes plotted as a function of ROI and sometimes as a function of conditions which is not ideal. As authors had no specific hypothesis on the localization of ERP responses as a function of the experimental condition I would suggest them to remove the ROI level from the ANOVA.

·           In a similar venue, it is not clear to me why spatial frequency was not introduced as a factor in the analysis. This would have allowed authors to make direct comparisons between ERP responses obtained in the LSF and HSF conditions. As opposed to ROI, this was something to which they had a hypothesis about.

·           In experiment 2 authors claim “Consistent with our expectation, the N170 showed enhancement to angry faces filtered by LSF. This result supports our hypothesis that rapid and coarse processing through magnocellular routes is involved in early threatening face differentiation”. However, this result emerged from the interaction between ROI and emotion being evident only across the right ROI and also being relatively close to the alpha threshold level of 0.05 (p = 0.042). Given that no specific hypothesis regarding the involvement of the right hemisphere in the processing of emotional faces was provided I don’t see the point once again in using different ROIs in the analysis.

·           While in experiment 1 the number of participants is justified by a power analysis this is not the case for experiment 2. Which was the point in increasing sample size of 26% (i.e., 7 participants)?

·           Why did the authors decide to remove the static condition in experiment 2?

·           Why not including ERP plot for experiment 2?

Comments on the Quality of English Language

Few missing words/verbs are present

Author Response

Reviewer 1

In the current study Yu and colleagues used EEG to determine “if the interactive threat effect produced by looming angry faces” reported in previous studies “is a product of a rapidly activating, coarse subcortical pathway”. To test this hypothesis, they decomposed face images in their LSF and HSF components. The idea is that, since signals from subcortical pathways rely on coarse information tuned to low spatial frequencies, the enhanced P1 response previously reported for angry looming faces should be evident over LSF stimuli and not on HSF ones. No evidence of such relationship was found as neither LSF nor HSF elicited an enhanced P1 response. Authors only confirmed (in Exp 2) previous results of larger responses for angry faces in BSF stimuli.

Although the topic of the study is interesting, I don’t consider it suitable for publication in Brain Sciences. Here are some of the motivations that led me to this choice alongside with some suggestions.

  • My main concern is about the design of the study which I consider not entirely suitable to answer the main question. I don’t see how authors could rule out the influence of the plethora of factors known to affect P1 amplitude while saying something very specific on the recruitment (or not) of visual subcortical routes. They conclude that their results suggest the “Existence of a rapid pathway for differentiating emotional expressions at the P1, but that this pathway is not reliant on LSF information”. First, the term “rapid” is ambiguous as seems to suggest the involvement of pathways other than the primary visual route. Second, this conclusion per se is also ambiguous as it should rather state “that this pathway is not reliant on LSF and HSF information in isolation” as authors found a null result even when using HSF stimuli.
  • Authors’ response: We appreciate these very relevant comments. The reviewer’s comments suggests to us that we have not expressed our thoughts clearly enough, and our word seem to overinterpret what our results are showing.

As we tried to make clear, our hypothesis was that the magnocellular pathway, running in parallel with the primary visual routes, is sufficiently rapid and can lead to the early modulation of looming angry faces on the P1 component. The magnocellular pathway may run subcortically through the superior colliculus pulvinar, as suggested by the evidence reviewed in the Introduction, (page 2, lines 65-89). Clearly, this is absolutely not the only route for motion processing, and after careful examination of the manuscript, we agree that we have not made this clear. We are grateful to the reviewer for bring this to our attention and have modified the manuscript accordingly, correcting our wording to make this point clearer (page 2, lines 83-87; page 3, lines 111-113; lines 124-129).

We also agree that the conclusion should have been more clearer. Indeed, we meant to make a point that the visual pathway is not reliant on either low or high spatial frequency bands in isolation. We have edited the manuscript now (page 1, line 19; page 17, lines 621-623)

Regarding the various factors that modulate the P1 component, we are of course in agreement that there are a number of features that produce modulations, particularly low-level characteristics or attention, as shown extensively in the literature. However, our current approach aimed to test the interaction of looming emotional stimuli irrespective of these other factors, by equating stimulus relevance (attention to the emotion) across conditions, and by factoring out the low-level features (which is also why we did not include the different frequency bands in a single ANOVA, see comment below). With the current experimental design, and under the assumption that low spatial frequency is preferably processed by the magnocellular pathway, our results suggest multiple routes for moving emotional stimuli. We have edited the Introduction to be clearer on the justifications (page3, lines 127-129)

  • The analysis of motion signals conveyed by subcortical routes are expected to reach the cortex at earlier latencies than those here explored (~50 ms). This has been extensively reported both on intracranial recording studies on monkeys (see for example Kawano et al., 1994), TMS studies (see for example Grasso et al. 2018; Laycock et al., 2007) and EEG studies on humans (see for example Ffytche et al., 1995). In the light of such literature, how would the authors claim that the enhanced P1 and N170 ERP responses reflect the recruitment of fast subcortical routes even in the case of a positive result on LSF stimuli?
  • Authors’ response: We thank the reviewer for this comment and for providing these references. As noted above, the aim of the study was to identify if and when emotion interacted with motion, and not to identify motion processing pathways alone. Consequently, our suggestion is not that the P1 or N170 are a fast subcortical pathway is the only reason for the early effects at the P1 and N170. We felt it was important to acknowledge the possibility of this magnocellular pathway being subcortically-based in the Introduction, since there is a considerable amount of literature supporting this notion. As a result, we related some of our discussion back to the subcortical vs cortical pathways for the early effects we found at the P1 and N170. Overall, our conclusion was in support of a multiplicity of visual pathways since the early processing stages. We have revised our wording to be more precise.
  • Furthermore, few studies in the literature directly investigated the visual pathways corresponding to looming/receding emotional faces. As a result, we lacked direct evidence pointing to the most possible neural networks. The current study was just the beginning, and we would always need to test this supposedly fast magnocellular pathway before moving on to other possibilities. We have edited the manuscript where appropriate to clarify this point, and included the suggested references (page 2, lines 74-78; page3, lines 127-129; page14, lines 465-466).

Cited References

Kawano, K., Shidara, M., Watanabe, Y., & Yamane, S. (1994). Neural activity in cortical area MST of alert monkey during ocular following responses. Journal of neurophysiology71(6), 2305-2324.

Grasso, P. A., Làdavas, E., Bertini, C., Caltabiano, S., Thut, G., & Morand, S. (2018). Decoupling of early V5 motion processing from visual awareness: A matter of velocity as revealed by transcranial magnetic stimulation. Journal of cognitive neuroscience30(10), 1517-1531.

Laycock, R., Crewther, D. P., Fitzgerald, P. B., & Crewther, S. G. (2007). Evidence for fast signals and later processing in human V1/V2 and V5/MT+: a TMS study of motion perception. Journal of Neurophysiology98(3), 1253-1262.

Ffytche, D. H., Guy, C. N., & Zeki, S. (1995). The parallel visual motion inputs into areas V1 and V5 of human cerebral cortex. Brain118(6), 1375-1394.

  •   I am not convinced about the statistical analyses used. In particular, different ERP components were analyzed using ANOVAs with a different number of categories within the level of ROI. In my view, it would be more straightforward to average across the nelectrodes eliciting the largest response regardless of their scalp localization. This would have reduced the number of levels while also making ANOVAs more comparable to each other. In addition, it would have also made ERPs plots much more comprehensible. Now ERPs are, sometimes plotted as a function of ROI and sometimes as a function of conditions which is not ideal. As authors had no specific hypothesis on the localization of ERP responses as a function of the experimental condition I would suggest them to remove the ROI level from the ANOVA.
  • Authors’ response: Thanks for pointing this out. After considering the reviewer’s suggestion, we agree that pooling one ROI for the P1 component would be a more straightforward way to present the results. We have rerun the relevant analyses for both our experiments, and the same findings were revealed as what we reported previously in the manuscript. It doesn’t change our understanding and discussion of the P1 results, but we have edited the manuscript where is relevant (page 6, lines 222-223 & 244-252; page 10, line 307-309; page 12, lines 388-389; page 13, lines 405-417). We have also modified the ERP traces for the P1 in experiment 1 to make it more comprehensive (page 8).
  • Regarding the N170, it is characterized by the strongest negativity around the bilateral temporal-occipital regions, and it is a common method to include left and right ROIs when analyzing the N170. Therefore, we felt the importance of having two ROIs for the N170 and including it as a factor in the ANOVAs.
  • The other comprehensive ERP traces of each component are now included in our supplementary materials. We realized there are many conditions regardless, and we would not want to distract the readers from the significant effects. Therefore, the N170 and P2 were plotted only to show the general trend and conditions with significant results. We have edited the manuscript to describe the methods more clearly (page 6, lines 238-239; page 13, lines 401-402)
  •  
  • In a similar venue, it is not clear to me why spatial frequency was not introduced as a factor in the analysis. This would have allowed authors to make direct comparisons between ERP responses obtained in the LSF and HSF conditions. As opposed to ROI, this was something to which they had a hypothesis about.
  • Authors’ response: As the reviewer pointed out above, early ERP responses are highly sensitive to low-level features. We thus feel that it is important to compare conditions within each frequency band to avoid such influences. Elsewhere, the literature has pointed out a differences in LSF vs HSF visual processing during early stages, and some studies using filtered stimuli by spatial frequency also chose to separate their analysis on each frequency band. For these reasons, we believe it is preferable not to include spatial frequency as a factor. We have edited the manuscript to justify this point clearly (page 3, lines 119, 126-130)
  • In experiment 2 authors claim “Consistent with our expectation, the N170 showed enhancement to angry faces filtered by LSF. This result supports our hypothesis that rapid and coarse processing through magnocellular routes is involved in early threatening face differentiation”.However, this result emerged from the interaction between ROI and emotion being evident only across the right ROI and also being relatively close to the alpha threshold level of 0.05 (p = 0.042). Given that no specific hypothesis regarding the involvement of the right hemisphere in the processing of emotional faces was provided I don’t see the point once again in using different ROIs in the analysis.
  • Authors’ response: Thanks for this comment. As we explained above, the N170 is characterised by a bilateral distribution and is typically analysed with the left and right ROIs as a factor. We still think having the two ROIs would be more sensitive to capture those milder effects. However, we appreciate that the reviewer pointed out the ambiguity in our hypothesis – it would be more precise to mention the potential ROI effect at the N170, as the literature has often reported asymmetrical modulation for the N170, and the emotional effect is usually more pronounced at the right hemisphere. We have edited the manuscript to accommodate this point (page 1, line 42)
  • While in experiment 1 the number of participants is justified by a power analysis this is not the case for experiment 2. Which was the point in increasing sample size of 26% (i.e., 7 participants)?
  • Authors’ response: The power analysis was meant to support the sufficiency of our sample size. Since the sufficiency is established for experiment 1, and the second experiment had more participants but the same number of factors (and one less level for the Motion), we didn’t think running the power analysis again for experiment 2 is necessary.
  • Why did the authors decide to remove the static condition in experiment 2?
  • Authors’ response: The static condition was initially introduced as a manipulation check to see if spatial frequency and motion interact during early visual processing (i.e., the P1 stage) of basic visual features. Experiment 1 showed no such effect at the P1. More importantly, experiment 2 changed the task from passive viewing to attending to facial expressions. To ensure the results were comparable with our previously published studies, including unfiltered faces is crucial. Therefore, we included broadband/BSF faces but removed static conditions to make the experiment more manageable. We have edited the methods in the manuscript to make this point clearer (page 3, line 119; page 11, lines 320-322 & lines 353-356)
  • Why not including ERP plot for experiment 2?
  • Authors’ response: We did not include this for the sake of clarity; we were worried that including more ERP plots would reduce the paper's readability and overwhelm the readers. Thus, we only stated in the manuscript that the ERP traces were in similar trends as those for Experiment 1. However, as the reviewer mentioned, we decided to include the plots in the supplementary materials for those interested. (we took another reviewer’s suggestion of publishing the appendix as supplementary material)

Reviewer 2 Report

Comments and Suggestions for Authors

The research goal, methods, and results are clearly described and presented.

I have one suggestion for section "2. Experiment 2".

Here, you firstly summarise again and propose interpretation of Experiment 1, to then introduce Exp 2 as an experiment to check for your explanation of exp 1's results. This way of presenting may be a bit confusing at a very first glance.

About Experiment 2 and the emotion decision task, I have the following question: did you check for the dominant hand used by participants and analysed potential delay in the Y or N key selection? Usually, in a decision task, participants have only two possible buttons in front of them, selected according to the dominant hand on the yes key.  

As a general suggestion, I would add the acknowledgment that the for both experiments the sample is small (n=26 for exp 1, n=30 for exp 2) ,although results are quite clear.

Author Response

The research goal, methods, and results are clearly described and presented.

I have one suggestion for section "2. Experiment 2".

Here, you firstly summarise again and propose interpretation of Experiment 1, to then introduce Exp 2 as an experiment to check for your explanation of exp 1's results. This way of presenting may be a bit confusing at a very first glance.

  • Authors’ response: Thanks for pointing out. We have modified the relevant wording to express more precisely (page 11, lines 318-324).

About Experiment 2 and the emotion decision task, I have the following question: did you check for the dominant hand used by participants and analysed potential delay in the Y or N key selection? Usually, in a decision task, participants have only two possible buttons in front of them, selected according to the dominant hand on the yes key.  

  • Authors’ response: Thank you for pointing this out. We have measured the handedness of each participant. However, we did not analyse this measure as we were only interested in their ERP results for this paper.

As a general suggestion, I would add the acknowledgment that the for both experiments the sample is small (n=26 for exp 1, n=30 for exp 2) ,although results are quite clear.

  • Authors’ response: Thank you for this suggestion. We have specified in the methods to acknowledge the sample sizes (page 3, line 147; page 11, lines 347-348).

Reviewer 3 Report

Comments and Suggestions for Authors

The paper entitled "Looming angry faces: differential electrophysiological dynamics for filtered stimuli by low and high spatial frequencies.", is very interesting, I think the range of applications could be very wide.

I have some suggestions for the authors:

1) In the method could be useful to insert a table with the data of the participants. 

2) the authors could insert a section with the statical analysis in detail

3) The Appendix A, could be published as supplementary material.

Comments on the Quality of English Language

Minor editing of English language required

Author Response

Reviewer 3

The paper entitled "Looming angry faces: differential electrophysiological dynamics for filtered stimuli by low and high spatial frequencies.", is very interesting, I think the range of applications could be very wide.

I have some suggestions for the authors:

1) In the method could be useful to insert a table with the data of the participants. 

- Authors’ response: Thank you for this suggestion. We considered making a table to detail the participants’ demographic information initially. However, we eventually shortened it by describing it in the text as the paper has become long. We have edited our method sections to make the information clearer (page 3, lines 141-142; page 11, lines 360-362).

2) the authors could insert a section with the statical analysis in detail

- Authors’ response: Thank you for this suggestion. We have tried to include comprehensive tables detailing the descriptive statistics of each ERP of each condition in the results sections. However, the tables seemed too large and made the reading a bit difficult to follow. We eventually decided to leave the table in the supplementary materials, but we improved the wording of the results in the manuscript.

3) The Appendix A, could be published as supplementary material.

- Authors’ response: Thank you for this suggestion. We have edited the manuscript to mention the supplementary material as well (page6, lines 236, 240; page 13, lines 399, 402)